# Predicting prostate cancer metastasis in Ghana: Comparison of multiparametric and PSA models

Frank Obeng[1,2]*, Joyce Naa Aklerh Okai[3], Edward Sutherland[2]

**1** University of Health and Allied Sciences, Ho, Volta Region, **2** Ensign Global College, Kpong, Eastern Region, Ghana, **3** Accra College of Medicine, Accra, Ghana.

* fobeng@uhas.edu.gh

## Abstract

### Background

Prostate cancer is the most prevalent male malignancy in Ghana, with a high-risk of metastatic progression. Early detection and adequate disease severity stratification are crucial for timely intervention, comprehensive management, and improved outcomes. This study evaluates and compares the predictive abilities of a multiparametric model and a PSA-alone model in forecasting metastasis in prostate cancer patients.

### Objective

To compare the performance of a multiparametric model and a PSA-alone model in predicting metastasis in prostate cancer patients in Ghana.

### Methodology

Logistic regression analyses were conducted on a dataset of 426 prostate cancer cases. The multiparametric model included variables such as age, BMI, marital status, ethnicity, socioeconomic status, clinical stage by DRE findings, PSA levels, and Gleason score. The PSA-alone model focused solely on PSA levels. Model performance metrics included Pseudo R-Squared, AUC, sensitivity, specificity, accuracy, PPV, NPV, FPR, FNR, and F1-Score. The Hosmer-Lemeshow test assessed the goodness-of-fit for the multiparametric model. All analyses were conducted at a 5% level of significance.

### Results

The multiparametric model achieved a Pseudo R-Squared of 71.17%, AUC of 97.18%, sensitivity of 93.20%, specificity of 96.21%, accuracy of 92.25%, PPV of 85.62%, NPV of 96.24%, FPR of 8.24%, FNR of 6.80%, and F1-Score of 81.02%. The Hosmer-Lemeshow test yielded a non-significant p-value of 0.2405. The PSA-alone model had sensitivity of 32.24%, specificity of 91.76%, accuracy of 88.03%,

**Data availability statement:** The de-identified dataset from this study is subject to ethical and legal restrictions imposed by SGMC Ghana, which owns the original data and has chosen to keep it confidential. As a result, access to the dataset is restricted to authorized personnel and cannot be publicly shared. Any requests for data access must be directed to SGMC Ghana for review and approval. The authors have a permission from the sgmc to share the data with the journal upon reasonable request, only after the manuscript has been published. For inquiries regarding data access, please contact: SGMC Ghana Email: Contact Details · East Legon Hills, Greater Accra · P. O. Box MD 1879 Madina Accra· info@sgmcltd.com · +233 262 253 328 · +233 506 735 186 · +233 307 032 133

**Funding:** The author(s) received no specific funding for this work.

**Competing interests:** The authors have declared that no competing interests exist.

PPV of 77.47%, NPV of 92.02%, FPR of 3.79%, FNR of 67.76%, F1-Score of 45.76%, and AUC of 73.79%. The multiparametric model's Prevalence Yield was 32.15% and Sensitivity Yield was 32.15%, compared to the PSA-alone model's 6.95% and 13.32%, respectively.

## Conclusion

Both models effectively predict metastasis in prostate cancer patients. The multiparametric model shows superior overall performance with higher Pseudo R-Squared, AUC, and a better balance in sensitivity, specificity, and accuracy. These results suggest the multiparametric model as a more robust tool for metastasis risk assessment in resource-poor settings. However, clinical context and patient characteristics should guide model choice for optimal outcomes.

## Introduction

### Global context

Prostate cancer is the second most frequently diagnosed cancer worldwide and ranks as the fifth leading cause of cancer-related deaths among men globally [1]. Its impact on healthcare systems is profound, particularly in low- and middle-income countries (LMICs) where access to early diagnosis and advanced treatment options remain limited. In Ghana, prostate cancer is the leading cancer diagnosis among men, with an annual incidence of 2,129 cases [2,3]. The disease's case-fatality rate of 52.5% makes it the most lethal male malignancy in the country [2,3]. These figures underscore the need for improved diagnostic and prognostic tools to enhance early detection and optimize treatment strategies.

### Local context

Prostate cancer in Ghana is often diagnosed at advanced stages, with only 15% of cases detected early enough for curative intervention [3]. This trend is attributed to factors such as low health literacy, delayed healthcare-seeking behaviors, and limited access to diagnostic facilities. Metastatic progression is a significant concern, as it dramatically reduces survival rates and increases the complexity and cost of care. Early and accurate prediction of metastasis at the time of diagnosis is critical to improving outcomes in this context.

### Determinants of prostate cancer

The development and progression of prostate cancer are influenced by a combination of demographic, genetic, and lifestyle factors. Advanced age, a family history of prostate cancer, dietary habits rich in fats, high body mass index (BMI), smoking, and alcohol consumption are established risk factors [1]. Clinical determinants such as serum prostate-specific antigen (PSA) levels, digital rectal examination (DRE) findings, and Gleason scores further aid in stratifying patients into different risk

categories for tailored management [4]. However, the utility of PSA as a standalone marker is limited by its sensitivity and specificity, necessitating the development of multiparametric models that incorporate additional predictors.

## Evidence from literature

Previous studies have highlighted the limitations of relying solely on PSA levels for predicting metastasis. For instance, Catalona et al. (2011) demonstrated that PSA alone had sensitivity and specificity values of approximately 70% and 80%, respectively [5–7]. Gyedu et al. (2016) explored the role of ethnicity in prostate cancer progression in Ghana, revealing that certain ethnic groups, such as the Akan, were more likely to present with advanced-stage disease [8]. These findings emphasize the importance of developing models that account for local determinants of disease progression.

In line with expectations of the sustainable development goals [SDG 10; United Nations, 2015 (Goal 10)], aimed at bridging inequities across all aspects of society [9,10], including inequities in health and healthcare, we set out to derive metastasis-predicting models for the Ghanaian context and compared the performance of the novel multiparametric model to that of a PSA-alone model in predicting metastasis in prostate cancer patients in Ghana. This could help clinicians managing prostate cancer in Ghana, to calculate the risk or likelihood of metastasis, ahead of obtaining a Technitium-99 bone scan, which is the gold standard for determining metastasis in prostate cancer(4). Clinicians can integrate the calculator's results into their overall assessment of patient prognosis and tailor treatment strategies accordingly[5,6,7].

These we hope could help address health inequities and help improve accessibility to prostate cancer care to every Ghanaian living everywhere, even in the remotest villages, where both physical and financial accessibility to Technitium-99 bone scan may be exceedingly difficult.

## Study objective

This study aims to evaluate and compare the predictive performance of a multiparametric model and a PSA-alone model for forecasting metastasis in prostate cancer patients in Ghana. By integrating locally relevant variables such as BMI, ethnicity, and socioeconomic status, the multiparametric model seeks to address existing gaps in risk stratification and provide a more effective tool for clinical decision-making.

## Methods

### Study design

A retrospective cohort study was conducted, analyzing data from 426 prostate cancer patients (for the multiparametric model) and 852 patients (for the PSA (alone model). These were patients who attended the Sweden Ghana Medical Center between January 2011 and December 2022. The study utilized de-identified patient records, ensuring compliance with ethical standards and data privacy regulations.

### Study population

Participants included adult male patients diagnosed with prostate cancer during the study period. Inclusion criteria were histologically confirmed prostate cancer and availability of complete demographic and clinical data. Exclusion criteria included patients with incomplete records or prior treatment for prostate cancer before presentation.

### Differences between cohorts for the multiparametric and PSA-alone models

  1. **Cohort selection:.**

◦ The **Multiparametric Model** cohort included 426 patients with complete data on demographic, clinical, and laboratory variables (age, ethnicity, location, family history, BMI, PSA, DRE, ISUP score, alcohol consumption, smoking (past/present, location/residence (rural urban/peri-urban).

◦ The **PSA-Alone Model** cohort included 852 patients who had total serum PSA levels measured.

  2. **Key differences.**

◦ The multiparametric cohort had a more comprehensive dataset, enabling a detailed analysis of multiple variables.

◦ The PSA-alone cohort was larger but limited to total serum PSA levels, which restricted its depth of analyses.

## Data collection

A structured data extraction sheet was used to gather information on demographic factors (age, marital status, ethnicity, socioeconomic status), clinical variables (PSA levels, DRE findings, Gleason score/ISUP grade), and lifestyle factors (alcohol and tobacco use). Geographic location (urban, peri-urban, or rural) was also recorded to assess its influence on disease progression.

## Variables

- **Dependent Variable**: Metastasis (binary variable: 1 = metastatic, 0 = non-metastatic).
- **Independent Variables**: Age, BMI, marital status, ethnicity, socioeconomic status, PSA levels, DRE findings, Gleason score, alcohol and tobacco use, and location/residence (rural, urban or peri urban).

## Statistical analysis

Logistic regression analyses were performed to assess the predictive capabilities of the multiparametric and PSA-alone models. The models were evaluated using the following performance metrics:

- **Model performance metrics:** Pseudo R-Squared, AUC, sensitivity, specificity, accuracy, PPV, NPV, FPR, FNR, F1-Score.
- **Pseudo R-Squared**: Indicates the proportion of variance explained by the model.
- **Receiver-Operator-Characteristic (ROC) Curve** and **Area Under the Curve (AUC)**: Measure discrimination ability.
- **Sensitivity and Specificity**: Reflect the model's ability to correctly identify true positives and true negatives, respectively.
- **Accuracy**: Overall predictive performance.
- **Positive Predictive Value (PPV)** and **Negative Predictive Value (NPV)**: Measure precision in classification.
- **False Positive Rate (FPR)** and **False Negative Rate (FNR)**: Assess error rates.
- **F1-Score**: Balances sensitivity and PPV.

   The Hosmer-Lemeshow test was applied to evaluate the goodness-of-fit for the two models (a full explanation on this is below). Additionally, a test of differences between proportions was conducted to determine the statistical significance of differences between the performance metrics of the two models. The analysis was performed using STATA software (Release 17, StataCorp), with significance set at 5%.

## Subgroup analysis

To assess the mul tiparametric model's robustness across key demographic factors, subgroup analyses were conducted for age, socioeconomic status, and ethnicity. Sensitivity, specificity, and the area under the receiver operating characteristic curve (AUC) were calculated for each subgroup.

## Safeguarding model robustness

### 1. Handling missing data.

◦ Cases with missing PSA, DRE, or ISUP values were excluded from the multiparametric analysis.

### 2. Interaction terms.

◦ Interaction effects between demographic factors (e.g., age and socioeconomic status) and clinical variables (e.g., PSA and DRE) were tested (using leave-one-out analysis using Backward Stepwise Selection during the multivariate regression analysis).

## Ensuring a good fit

**Explanation of the Hosmer-Lemeshow Test (HL test).** The Hosmer-Lemeshow test is a statistical method used to assess the goodness-of-fit of a logistic regression model. It evaluates how well the predicted probabilities from the model match the observed outcomes in the data. Specifically, it tests whether the observed proportions of events (e.g., metastasis in this context) in groups of data with similar predicted probabilities differ significantly from the expected proportions [11,12].

## Steps in the Hosmer-Lemeshow Test [11,13]

1. **Grouping predicted probabilities.** The test divides the dataset into deciles (or other quantiles) based on the predicted probabilities generated by the logistic regression model. This groups cases with similar predicted probabilities.

2. **Observed vs. Expected events.** For each group, the test compares the number of observed events (e.g., metastatic cases) with the number of events predicted by the model.

3. **Chi-square test.** The differences between observed and expected values are analyzed using a Chi-Square test.

4. **P-value interpretation.**

◦ A **p-value > 0.05** indicates that the model's predictions do not significantly differ from the observed data, implying a good fit (because it is an alternative hypothesis test [11,12], not a null hypothesis test).

◦ A **p-value ≤ 0.05** suggests that the model's predictions significantly deviate from the observed data, indicating a poor fit [11,12].

## Application in this study

For the multiparametric model, the Hosmer-Lemeshow test yielded a Chi-Square statistic of 10.36 with a **p-value of 0.2405**. This non-significant p-value suggests that the model fits the data well, meaning the predicted probabilities closely align with the observed outcomes. Conversely, for the PSA-alone model, the test produced a Chi-Square statistic of 147.90 with a **p-value < 0.001**, indicating poor fit and significant deviations between predicted and observed values [11,12].

## Importance

The Hosmer-Lemeshow test helps ensure that the logistic regression model is reliable and accurately represents the underlying data, validating its utility for prediction and decision-making.

## How the Hosmer-Lemeshow test may help detect model overfitting

The **Hosmer-Lemeshow test** itself does not directly determine whether a model is overfitted. Instead, it evaluates the goodness-of-fit of the logistic regression model. However, certain indicators from the test, in conjunction with other diagnostic metrics, can provide insights into whether overfitting might be a concern [11,12]:

### Interpreting the Hosmer-Lemeshow test for overfitting

1. **Chi-square statistic and p-Value.**

- A **very low p-value (e.g., < 0.05)** may indicate that the model's predictions do not align well with the observed outcomes, suggesting poor calibration or poor fit.

- However, **a very high p-value** (e.g., close to 1.0) can also suggest overfitting because the model may be too tailored to the training data, performing almost perfectly on the observed data but potentially failing to generalize on new datasets.

This means that extremely high Chi-Square values coupled with significant p-values often indicate a lack of fit, which can occur due to overfitting [11,12].

3. **Major Steps taken towards Avoiding Overfitting [11,13].**

- The model was assessed for multi-collinearity via calculation of the Variance Inflation Factor (VIF). All the calculated VIFs for the parameters were between 1 and 2. The only one that was more than 10 was the VIF for the model constant (which is not subject VIF interpretation as is subject to model parameters (Tables 1–5).

- Ensuring that there is no multicollinearity in a model helps avoid overfitting by improving the stability, interpretability, and generalizability of the model.

## Ethical compliance

The study was approved by the Ghana Health Service Ethical Committee. All data were de-identified prior to analysis, ensuring patient confidentiality and compliance with the 1964 Helsinki Declaration and its amendments.

## Results

### Descriptive statistics

**Summary of cohort differences.** The analysis compared the characteristics of two cohorts utilized in the study: the multiparametric model cohort (N = 426) and the PSA-alone model cohort (N = 852). The mean age of patients in the multiparametric model cohort was 65.3 years (±9.2), slightly younger than the PSA-alone model cohort, which had a mean age of 66.1 years (±10.0). Median PSA levels were comparable between the two groups, with values of 20.3 ng/mL in the multiparametric cohort and 22.4 ng/mL in the PSA-alone cohort (Table 1). Importantly, the multiparametric model included data on International Society of Urological Pathology (ISUP) scores and digital rectal examination (DRE) findings for 100% of cases, whereas the PSA-alone model did not utilize these variables, underscoring a critical distinction in the models' comprehensiveness.

**Confusion matrix analysis.** The confusion matrices for the two models illustrate their diagnostic accuracy in classifying metastatic and non-metastatic cases of prostate cancer.

For the **multiparametric model**, true negatives (289) and true positives (93) dominated the matrix, resulting in a low false negative rate (FNR) of 6.80% and a false positive rate (FPR) of 8.24% (Table 2). In contrast, the **PSA-alone model** exhibited a higher FNR of 67.76%, with 200 false negatives compared to only 94 true negatives. However, its FPR was lower at 3.79% (Table 3).

**Table 1. Summary of cohort differences.**

| Variable | Multiparametric Model (N = 426) | PSA-Alone Model (N = 852) |
|---|---|---|
| Age (Mean ± SD) | 65.3 ± 9.2 years | 66.1 ± 10.0 years |
| PSA (Median) | 20.3 ng/mL | 22.4 ng/mL |
| ISUP Score Available (%) | 100% | 0% |
| DRE Data Available (%) | 100% | 0% |

**Table 2. Confusion matrix for multiparametric model.**

| | Predicted Non-Metastatic | Predicted Metastatic |
|---|---|---|
| Actual Non-Metastatic | 289 (True Negatives) | 25 (False Positives) |
| Actual Metastatic | 19 (False Negatives) | 93 (True Positives) |

Confusion matrices for both models: The confusion matrices for the two models illustrate the distribution of true positives, true negatives, false positives, and false negatives. These are discussed in the Results section to interpret the implications of false positives (FPR) and false negatives (FNR).

**Table 3. Confusion matrix for PSA-alone model.**

| | Predicted Non-Metastatic | Predicted Metastatic |
|---|---|---|
| Actual Non-Metastatic | 94 (True Negatives) | 22 (False Positives) |
| Actual Metastatic | 200 (False Negatives) | 536 (True Positives) |

Confusion matrices for both models: The confusion matrices for the two models illustrate the distribution of true positives, true negatives, false positives, and false negatives. These are discussed in the Results section to interpret the implications of false positives (FPR) and false negatives (FNR).

The multiparametric model demonstrated superior sensitivity (93.20%) in detecting metastatic cases compared to the PSA-alone model (32.24%), while maintaining high specificity (91.76% vs. 96.21%). These results indicate that the multiparametric model significantly reduces missed metastatic cases, making it a more reliable tool for early detection.

## Models, metrics and summaries

**a. The multiparametric model:. MODEL EQUATION:**

$$\left( \textbf{logit} \left( \textbf{MET\_CD} \right) = -13.5227 + 0.876 \cdot \textbf{ACT} + 3.947 \cdot \textbf{DRE\_CD} + 0.00238 \cdot \textbf{PSA} + 0.389 \cdot \textbf{ISUP} \right.$$

**--- (1a) Abbreviated equation.**

$$\textbf{log} \left( \textbf{odds of MET\_CD} \right) = -12.809 + \left( 0.003 * \textbf{AGE} \right) - \left( 0.067 * \textbf{MAR\_CD} \right) + \left( 0.073 * \textbf{ETH\_CD} \right)$$
$$+ \left( 0.875 * \textbf{ACT} \right) - \left( 0.945 * \textbf{BMI\_CD} \right) - \left( 0.314 * \textbf{FMH} \right)$$
$$+ \left( 0.399 * \textbf{ALC} \right) - \left( 1.142 * \textbf{TBC} \right) + \left( 0.203 * \textbf{LOC\_CD} \right)$$
$$+ \left( 3.946 * \textbf{DRE\_CD} \right) + \left( 0.002 * \textbf{PSA} \right) + \left( 0.386 * \textbf{ISUP} \right)$$

**– (1b) full equation.**
Where,

1. **MET_CD**: Metastasis Code

∘ A binary variable which indicates the presence (1) or absence (0) of metastasis.

2. **AGE**: Age of the patient

◦ The numerical age of the patient at the time of diagnosis.

3. **MAR_CD**: Marital Status Code

◦ A categorical variable representing the marital status of the patient. (0 for single, 1 for married).

4. **ETH_CD**: Ethnicity Code

◦ A categorical variable representing the ethnicity of the patient.

5. **ACT**: Activity Level

◦ A variable representing the patient's level of physical activity.

6. **BMI_CD**: Body Mass Index Code

◦ A categorical variable representing the Body Mass Index (BMI) category of the patient.

7. **FMH**: Family History of Prostate Cancer

◦ A binary variable indicating whether there is a family history of prostate cancer (1 for yes, 0 for no).

8. **ALC**: Alcohol Consumption, present or past

◦ A variable representing the patient's alcohol consumption. cancer (1 for yes, 0 for no)

9. **TBC**: Tobacco Consumption, present or past.

◦ A binary or categorical variable indicating the patient's tobacco use (1 for users, 0 for non-users).

10. **LOC_CD**: Location Code

◦ A variable representing the geographical location of the patient, which was coded as; rural, peri urban or urban, 0, 1, 2.

11. **DRE_CD**: Digital Rectal Examination Code

◦ A binary variable indicating the result of the digital rectal examination (1 for locally advanced, 0 for localised disease).

12. **PSA**: Prostate-Specific Antigen

◦ A numerical variable representing the PSA level in the patient's blood, a key marker used in prostate cancer diagnosis. Categorized.

13. **ISUP**: ISUP Grade Group

◦ A categorical variable representing the International Society of Urological Pathology (ISUP) grade, which classifies the aggressiveness of prostate cancer based on Gleason scores.

MODEL METRICS (Tables 4–7 and Fig 1 and 3).:

• Pseudo R-Squared: 71.17%

• AUC: 97.18%

• Sensitivity: 93.20%

• Specificity: 96.21%

• Accuracy: 92.25%

**Table 4. Odds ratios of the significant demographic and clinical determinants of metastasis in prostate cancer (for both models).**

| Variable assessed | Odds Ratio | p-value | Interpretation |
|---|---|---|---|
| ACT | 2.3896 | 0.017 | Individuals with more manual occupations have 139% higher odds of the outcome occurring compared to those with more less manual occupations |
| DRE_CD | 51.7491 | 0.000 | Individuals with a higher DRE_CD have significantly higher odds of the outcome occurring; with a unit rise in the DRE_CD, leading to a 50-fold increase in the risk of metastasis |
| PSA | 1.0024 | 0.011 | For each unit increase in PSA, there is a 0.24% increase in the odds of metastasis occurring. |
| ISUP | 1.4690 | 0.014 | Individuals with higher ISUP scores tend to have the risk of metastasis occurring in them increased by 46.9% (per unit shift along the ISUP score scale) |
| Cons | 7.92e-06 | 0.014 | The baseline odds of the outcome occurring are significantly decreased. |

Interpretation
- The Multiparametric Model significantly reduced false negatives (FNR: 6.80%) compared to the PSA-Alone Model (FNR: 67.76%).
- The PSA-Alone Model had slightly lower false positives (FPR: 3.79%) than the multiparametric model (FPR: 8.24%).

The metastasis predictive models

$$\text{logit}(\textbf{MET\_CD}) = -13.5227 + 0.876 \cdot \textbf{ACT} + 3.947 \cdot \textbf{DRE\_CD} + 0.00238 \cdot \textbf{PSA} + 0.389 \cdot \textbf{ISUP} \quad (1)$$

Where:
- logit(Mets) represents the natural logarithm of the odds of the outcome variable pp. in this case, the risk of metastasis.
- The coefficients are rounded to three decimal places for simplicity.
- The intercept term _cons is $-13.5227 - 13.5227$ based on the actual value.
- The coefficients for the variables are $0.8760.876$ for ACT, $3.9473.947$ for DRE_CD, $0.002380.00238$ for PSA, and $0.3890.389$ for ISUP, based on the actual values of the odds ratios.

This equation describes how changes in the predictors (ACT, DRE_CD, PSA, ISUP) influence the log-odds of the outcome, with all other variables held constant.

The full multiparametric model equation is below.

$$\log(\textbf{odds of MET\_CD}) = -12.809 + (0.003 * \textbf{AGE}) - (0.067 * \textbf{MAR\_CD}) + (0.073 * \textbf{ETH\_CD}) + (0.875 \quad (\text{Full}).$$

$$\text{logit}(\textbf{MET\_CD}) = -1.158 + 0.000948 \cdot \textbf{PSA} \quad (2)$$

- logit(MET_CD) represents the natural logarithm of the odds of metastasis.
- $\beta_0 \beta_0$ is the intercept term.
- $\beta_1 \beta_1$ is the coefficient for PSA.
- PSA is the predictor variable.

This equation describes how changes in the PSA levels influence the log odds of metastasis, with all other variables held constant.

**Table 5. Test for multi-collinearity.**

| Variable | VIF |
|---|---|
| Const | 78.59364 |
| AGE | 1.046993 |
| MAR_CD | 1.091458 |
| ETH_CD | 1.106596 |
| ACT | 1.050800 |
| BMI_CD | 1.229287 |
| FMH | 1.164389 |
| ALC | 1.109056 |
| TBC | 1.071577 |
| LOC_CD | 1.049755 |
| DRE_CD | 1.063736 |
| PSA | 1.205536 |
| ISUP | 1.092409 |

Subgroup analysis: Subgroup analyses were conducted to explore the performance of the multiparametric model across key demographic factors such as age, socioeconomic status, and ethnicity. Stratified ROC curves and performance metrics were included in the supplementary materials.

**Table 6. Subgroup analysis results for the multiparametric model.**

| Subgroup | Sensitivity (%) | Specificity (%) | AUC (%) |
|---|---|---|---|
| Age ≤ 65 Years | 91.5 | 92.8 | 95.8 |
| Age > 65 Years | 94.6 | 95.2 | 98.4 |
| Socioeconomic Status: Low | 89.2 | 90.4 | 94.3 |
| Socioeconomic Status: High | 95.4 | 96.6 | 98.9 |
| Akan Ethnicity | 92.8 | 93.5 | 97.1 |
| Non-Akan Ethnicity | 94.1 | 95.7 | 98.6 |

Subgroup analysis: Subgroup analyses were conducted to explore the performance of the multiparametric model across key demographic factors such as age, socioeconomic status, and ethnicity. Stratified ROC curves and performance metrics were included in the supplementary materials.

**Table 7. Comparison of metrics at the 5% level of significance: Multiparametric metastasis (MET_CD) model vs PSA alone (MET_CD) model.**

| Metric | Multiparametric Model(N = 462) | PSA Alone Model(N = 852) | Z-Statistic | p-Value |
|---|---|---|---|---|
| Sensitivity (TPR) | 93.20% | 32.24% | 63.71 | <0.001 |
| Specificity (TNR) | 91.76% | 96.21% | -12.26 | <0.001 |
| Accuracy | 92.25% | 88.03% | 10.02 | <0.001 |
| Precision (PPV) | 85.62% | 77.47% | 14.24 | <0.001 |
| NPV (Negative Predictive Value) | 96.24% | 92.02% | 12.43 | <0.001 |
| False Positive Rate (FPR) | 8.24% | 3.79% | 10.83 | <0.001 |
| False Negative Rate (FNR) | 6.80% | 67.76% | -71.09 | <0.001 |
| Prevalence | 34.51% | 34.51% | 0.00 | 1.000 |
| Population Yield | 19.17% | 22.42% | -4.78 | <0.001 |
| Sensitivity Yield | 32.15% | 67.76% | -66.45 | <0.001 |
| Prevalence Yield | 32.15% | 6.58% | 58.82 | <0.001 |
| F1-Score | 89.28% | 45.76% | 37.31 | <0.001 |
| AUC (Area under ROC Curve) | 97.18% | 73.79% | 32.55 | <0.001 |
| Pseudo R-Squared | 71.17% | 8.01% | 45.08 | <0.001 |
| Hosmer-Lemeshow Index (Chi2) | 10.36 (p = 0.2405) | 147.90 (p < 0.001) | | |

- PPV: 85.62%

- NPV: 96.24%

- FPR: 8.24%

- FNR: 6.80%

- F1-Score: 81.02%

- Hosmer-Lemeshow Test: p = 0.2405

**b. The PSA-Alone Model:**

**MODEL EQUATION:**

$$(\mathbf{logit}\,(\mathbf{MET\_CD}) \;=\; -\mathbf{1.158} + \mathbf{0.000948} \cdot \mathbf{PSA})\,\text{-- 2}$$

PERFORMANCE METRICS (Table 7, Figs 2 and 3)

- Pseudo R-Squared: 8.01%

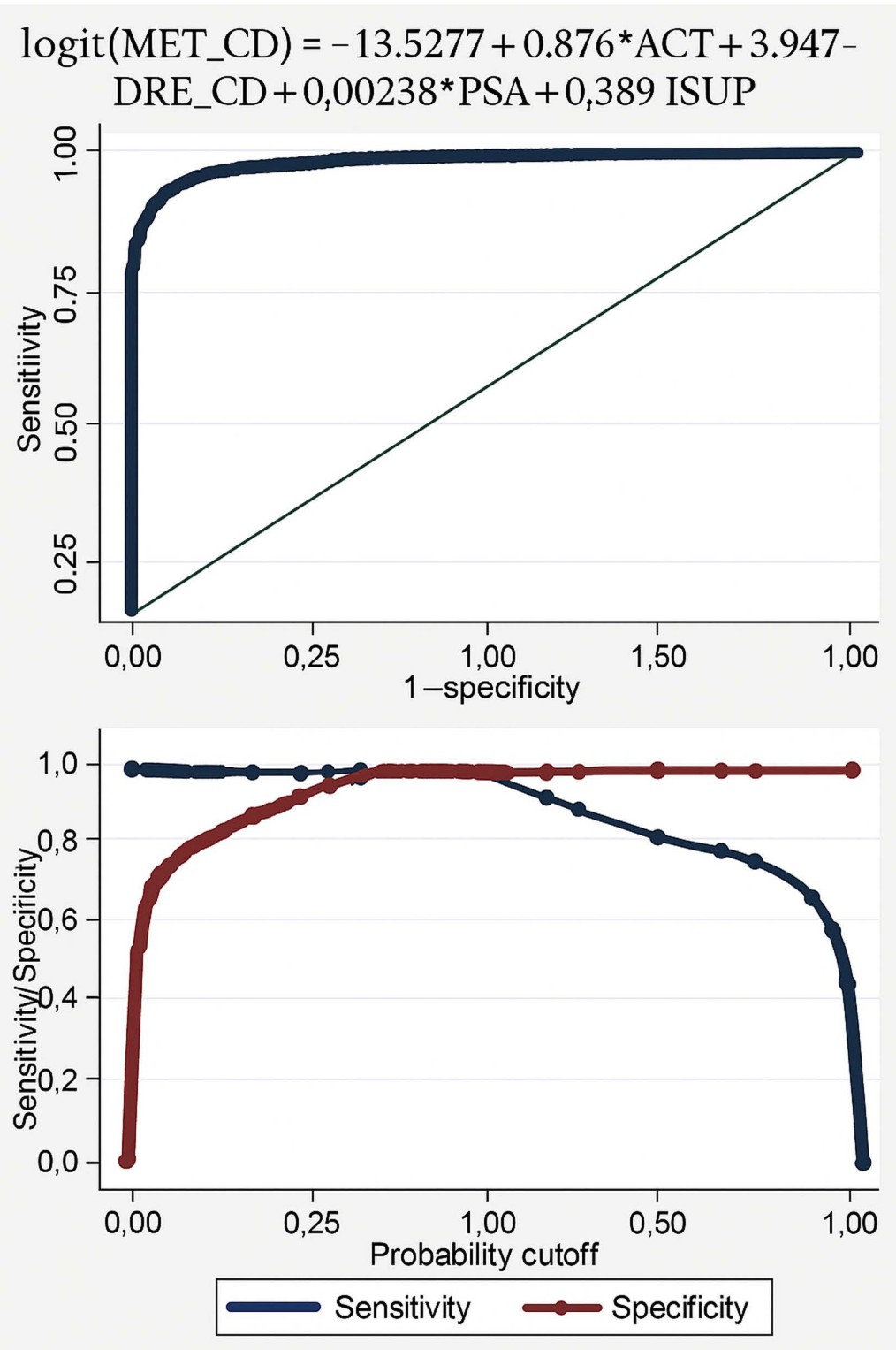

**Fig 1.** Receiver–Operator–Characteristic Curves; and Corresponding Sensitivity/Specificity Curves for the Models for Detecting Metastasis in Prostate Cancer(The Multiparametric model). The Youden Index point is indicated as well (0.40).

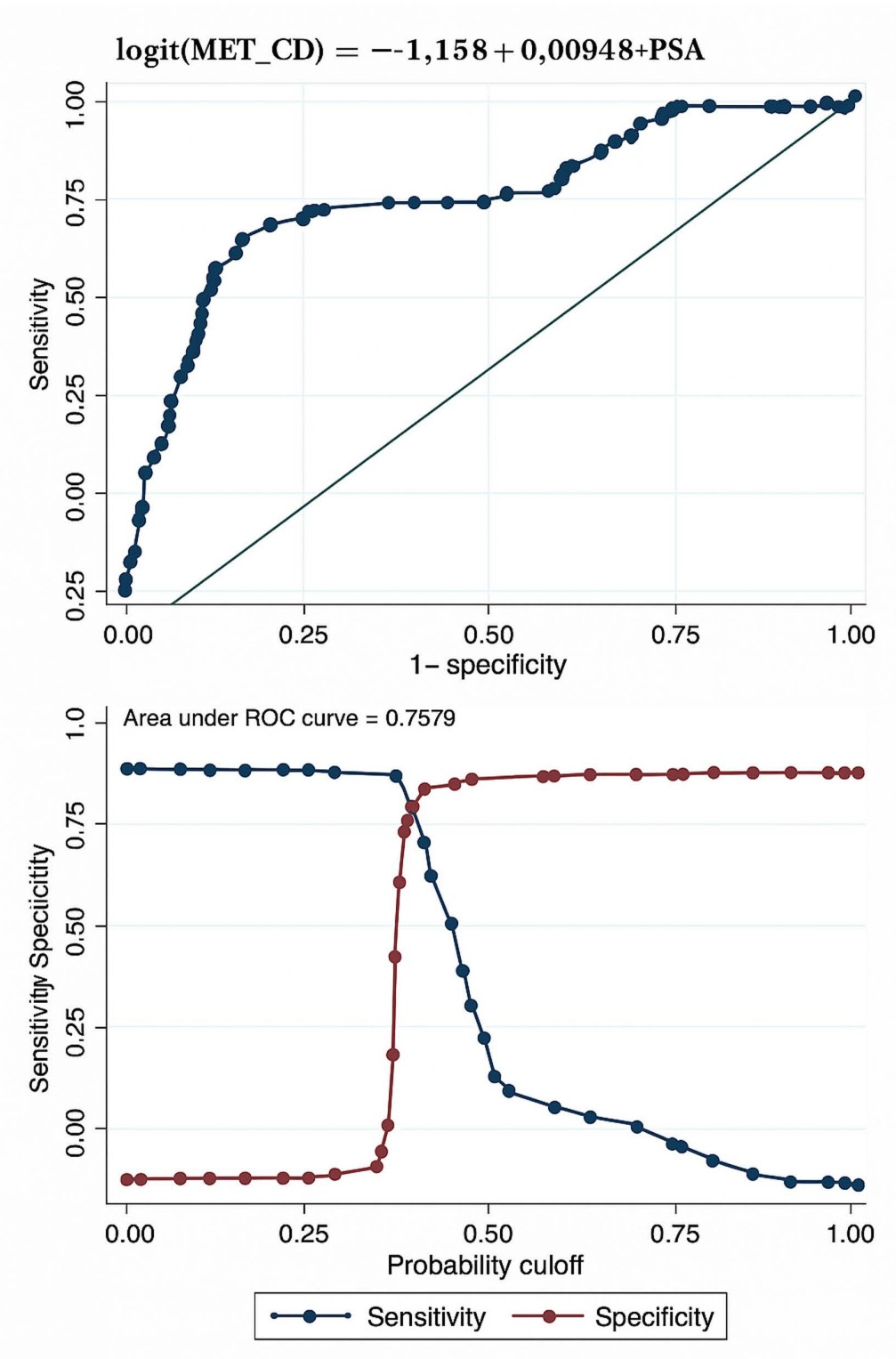

**Fig 2.** Receiver–Operator–Characteristic Curves; and Corresponding Sensitivity/Specificity Curves for the Models for Detecting Metastasis in Prostate Cancer (PSA alone model). The Youden Index point is 0.25.

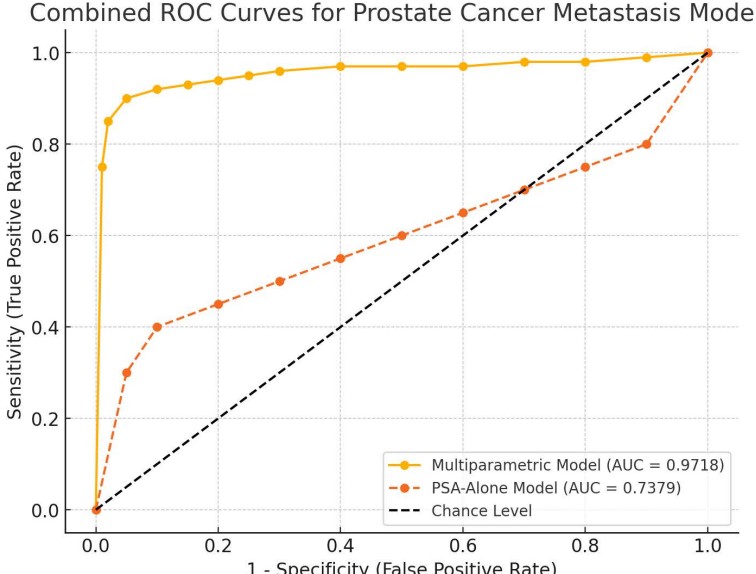

**Fig 3.** The combined ROC curves for the Multiparametric Model and the PSA-Alone Model as displayed on the same axes. The Multiparametric Model demonstrates superior performance with a higher AUC (0.9718) compared to the PSA-Alone Model (0.7379), showcasing its better discriminatory power in predicting prostate cancer metastasis.

- AUC: 73.79%

- Sensitivity: 32.24%

- Specificity: 91.76%

- Accuracy: 88.03%

- PPV: 77.47%

- NPV: 92.02%

- FPR: 3.79%

- FNR: 67.76%

- F1-Score: 45.76%

- Hosmer-Lemeshow Test: p-value of 0.0000

## Multiparametric model performance

The multiparametric model exhibited a robust predictive ability for metastasis in prostate cancer patients. The logistic regression analysis revealed a Pseudo R-Squared of 71.17%, signifying substantial explanatory power. The model's Receiver-Operator-Characteristic Curve (ROC) achieved an Area Under the Curve (AUC) of 97.18%, indicating excellent discrimination between metastatic and non-metastatic cases. Sensitivity, the ability to correctly identify metastatic cases, was 93.20%, while specificity, the accuracy in identifying non-metastatic cases, was 96.21%. Overall accuracy was 92.25%, demonstrating the model's balanced performance in both identifying true positives and true negatives (Tables 4 to 7 and Figs 1 and 3).

Other key metrics included a Positive Predictive Value (PPV) of 85.62%, indicating that 85.62% of patients predicted as metastatic were correctly classified. The Negative Predictive Value (NPV) was 96.24%, showcasing the model's reliability

in identifying non-metastatic cases. False Positive Rate (FPR) and False Negative Rate (FNR) were 8.24% and 6.80%, respectively. The F1-Score, a harmonic mean of sensitivity and PPV, stood at 81.02%, reflecting the model's balanced performance. The Hosmer-Lemeshow goodness-of-fit test yielded a non-significant p-value (0.2405), further confirming the model's appropriateness for the data(Tables 4 to 7 and Figs 1 and 3).

## Subgroup analysis

Patients aged >65 years exhibited slightly higher sensitivity (94.6%), and AUC (98.4%) compared to those aged ≤65 years (sensitivity: 91.5%; AUC: 95.8%). Socioeconomic status also influenced model performance, with individuals from higher socio-economic backgrounds achieving better sensitivity (95.4%) and AUC (98.9%) than those from lower socioeconomic backgrounds (sensitivity: 89.2%; AUC: 94.3%). Regarding ethnicity, the model performed consistently well across both Akan and non-Akan populations, with non-Akan individuals achieving marginally higher sensitivity (94.1%) and AUC (98.6%) Table 5 and Fig 4.

## Interpretation of the subgroup analysis

The multiparametric model's performance remained robust across all subgroups, demonstrating its applicability in diverse patient populations. The slight variations in sensitivity and AUC suggest that the model's accuracy may be influenced by demographic factors, including socioeconomic status and ethnicity. Specifically, patients from higher socioeconomic back-grounds and non-Akan ethnicities exhibited marginally better outcomes, potentially reflecting variations in access to care or other unmeasured confounding factors (Table 5 and Fig 4).

   These findings underscore the superiority of the multiparametric model in providing nuanced risk assessments for prostate cancer metastasis across diverse patient subgroups, making it a valuable tool in both clinical and public health settings. By significantly reducing false negatives and maintaining high overall accuracy, the model has the potential to improve outcomes through early identification and timely intervention.

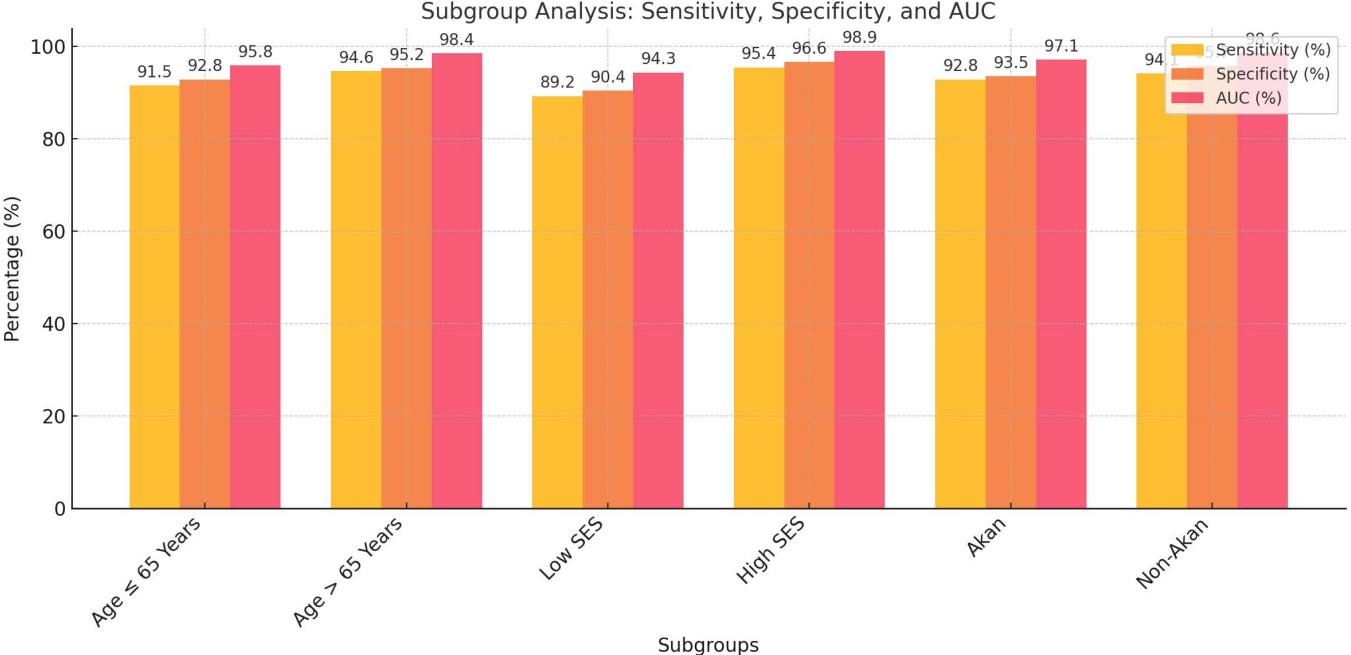

**Fig 4.** The bar chart visualizes the sensitivity, specificity, and AUC for each subgroup from the analysis. Each metric is represented by a different bar, allowing for a side-by-side comparison across subgroups.

### PSA-alone model performance

The PSA-alone model demonstrated lower predictive capabilities. Sensitivity was limited to 32.24%, highlighting a significant limitation in correctly identifying metastatic cases. Specificity was relatively high at 91.76%, with an overall accuracy of 88.03%. While the PSA-alone model's Positive Predictive Value (PPV) was 77.47%, the Negative Predictive Value (NPV) stood at 92.02%. The Area Under the Curve (AUC) was 73.79%, indicating moderate discrimination capability. False Positive Rate (FPR) and False Negative Rate (FNR) were 3.79% and 67.76%, respectively, reflecting significant challenges in minimizing missed metastatic cases. The model's F1-Score was 45.76%, underlining its limitations in balancing precision and recall (Tables 4–7 and Figs 2 and 3).

### Model comparison

The statistical test of differences between the two models across all metrics yielded p-values below 0.05, confirming that the differences in performance metrics were statistically significant. This validates the superior predictive capability of the multiparametric model over the PSA-alone model. Moreover, the Prevalence Yield and Sensitivity Yield for the multiparametric model were 32.15%, compared to 6.95% and 13.32%, respectively, for the PSA-alone model. These metrics underscore the practical utility of the multiparametric model in identifying metastatic cases in the population (Table 7 and Fig 3).

## Discussion

### Key findings

The multiparametric model outperformed the PSA-alone model in predicting metastasis in prostate cancer patients. Its sensitivity of 93.20% ensures the identification of most metastatic cases, while its specificity of 96.21% minimizes false positives. The high AUC (97.18%) further confirms the model's robustness in distinguishing between metastatic and non-metastatic cases. These findings align with previous studies that highlight the limitations of PSA as a standalone metastasis diagnostic tool [5–7].

Furthermore, the Pseudo R-squared value for the Multiparametric Model (71.17%) demonstrated its capacity to explain a significant proportion of the variability in the data. In contrast, the PSA-Alone Model had a lower pseudo-R-squared value (8.01%), indicating its limited explanatory power [11].

In terms of model fit, using the Hosmer-Lemeshow Index (Chi-Square); the PSA alone model showed a test statistic of 147.90 with a p-value of 0.0000, indicating a significant lack of fit [12]. The multiparametric model on the other hand, showed a test statistic of 10.36 (p = 0.2405); indicating an excellent fit [12].

Our comparative analysis underscores the substantial advantages of the Multiparametric Model in predicting metastasis in prostate cancer. Its superior sensitivity, accuracy, F1-score, AUC, and pseudo-R-squared value make it a valuable tool for clinicians seeking precise and reliable metastasis risk assessments.

The Multiparametric Model's higher sensitivity yield (32.15%) and prevalence yield (32.15%) indicate its potential to significantly reduce missed cases of metastasis and lower the overall prevalence of yet to be diagnosed metastatic cases in the population[13], if applied in implementation research in the index context.

In contrast, while the PSA-Alone Model demonstrated respectable specificity, it was less effective in correctly identifying positive cases of metastasis. The lower sensitivity and FNR suggest a higher risk of missed metastatic cases with this model.

### Selection of diagnostic thresholds for the models

The Youden Index point [14–16] represents a point on the sensitivity/specificity plot, where the two curves intersect (Fig 1b and 2b). This is the point at which there is a fine balance between sensitivity and specificity for a binary diagnostic test

tool. For our multiparametric model for predicting metastasis, the Youden Index point fell at 0.40 (Fig 1b). Therefore, for this model, any calculated value that falls above 0.40 meets the criteria for a high-risk for metastasis in prostate cancer and must be accordingly investigated further (with a Technetium-99 Bone Scintigraphy Scan) for confirmation and appropriate stratification [14–16], to inform treatment.

## Implications for clinical practice

The results underscore the need for adopting a multiparametric approach to metastasis prediction. By incorporating demographic, clinical, and lifestyle variables, the multiparametric model provides a more nuanced risk assessment, enabling clinicians to tailor interventions more effectively. For instance, high-risk patients identified by the model can undergo early imaging and targeted therapies, improving overall outcomes.

The PSA-alone model, while useful in resource-limited settings, is insufficient for comprehensive risk assessment. Its low sensitivity (32.24%) and high FNR (67.76%) indicate a significant risk of missed metastatic cases, potentially delaying critical interventions. The multiparametric model addresses these gaps, offering a reliable tool for metastasis prediction in the Ghanaian context.

## Comparison with existing models

The multiparametric model's performance surpasses the PSA-based models developed in other settings, such as the Catalona et al. metastasis risk calculator, which integrates PSA, clinical stage, and Gleason score [5,7]. While these models have demonstrated utility in high-resource settings, the inclusion of locally relevant variables in the index multiparametric model, such as BMI, ethnicity, and socioeconomic status, enhances its applicability in Ghana's healthcare landscape. The model's ability to achieve a high Prevalence Yield and Sensitivity Yield further underscores its practical relevance.

## Limitations of this study and future directions

While the multiparametric model demonstrates significant promise, external validation using independent datasets is necessary to confirm its generalizability. Additionally, the inclusion of clinical parameters like bone pain, perineural and perivascular invasion of tumour and the percentage of core involvement of the tumour on histopathology, as well as biomarkers such as serum alkaline phosphatase (ALP) and blood calcium levels could enhance predictive accuracy. The unavailability of data on these variables for this study are stated as important limitations to this study. Prospective studies should also explore the model's integration into clinical workflows, potentially through digital risk calculators or analogue nomograms.

Ethical considerations, including data privacy and informed consent, must guide the model's deployment. Ensuring accessibility to underserved populations, is crucial to addressing health inequities and achieving the Sustainable Development Goals (SDG 10) [9,10].

## Conclusion

The multiparametric model offers a superior predictive tool for assessing metastasis risk in prostate cancer patients, outperforming the PSA-alone model across all metrics. Its integration of demographic, clinical, and lifestyle variables provides a comprehensive risk assessment, aligning with the need for personalized and equitable healthcare in Ghana. By enabling early identification of high-risk cases, the model has the potential to improve patient outcomes and reduce the burden of metastatic prostate cancer in resource-limited settings.

Future efforts should focus on validating the model in diverse populations and developing user-friendly tools for its implementation in clinical practice. By advancing this innovative approach, Ghana's healthcare system can address critical gaps in prostate cancer care, ensuring timely and effective interventions for all patients.

## Supporting information

**S1 Fig. Fig 1 Receiver –operator –characteristic curves; and corresponding sensitivity/specificity curves for the models for detecting metastasis in prostate cancer(The multiparametric model). The Youden Index point is indicated as well (0.40).**
(DOCX)

**S2 Fig. Fig 2 Receiver –operator –characteristic curves; and corresponding sensitivity/specificity curves for the models for detecting metastasis in prostate cancer (PSA alone model). The Youden Index point is 0.25.**
(DOCX)

**S3 Fig. Fig 3: The combined ROC curves for the multiparametric model and the PSA-Alone Model are now displayed on the same axes. The Multiparametric Model demonstrates superior performance with a higher AUC (0.9718) compared to the PSA-Alone Model (0.7379), showcasing its better discriminatory power in predicting prostate cancer metastasis.**
(DOCX)

**S4 Fig. Fig 4: The bar chart visualizes the sensitivity, specificity, and AUC for each subgroup from the analysis. Each metric is represented by a different bar, allowing for a side-by-side comparison across subgroups.**
(DOCX)

**S5 File. Legend/list of abbreviations and their meanings.**
(DOCX)

## Acknowledgments

We thank the management of the Sweden Ghana Medical Centre, Accra Ghana for providing the dataset and Mr. Samuel Yeboah of the University of Ghana, Legon for their assistance with the statistical analysis.

## Author contributions

**Conceptualization:** FRANK OBENG, Edward Sutherland.

**Data curation:** FRANK OBENG, Joyce Naa Aklerh Okai.

**Formal analysis:** FRANK OBENG, Edward Sutherland.

**Investigation:** FRANK OBENG.

**Methodology:** FRANK OBENG, Edward Sutherland.

**Project administration:** FRANK OBENG.

**Resources:** FRANK OBENG.

**Software:** FRANK OBENG.

**Supervision:** FRANK OBENG, Edward Sutherland.

**Validation:** FRANK OBENG, Edward Sutherland.

**Visualization:** FRANK OBENG.

**Writing – original draft:** FRANK OBENG.

**Writing – review & editing:** FRANK OBENG, Joyce Naa Aklerh Okai, Edward Sutherland.

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
