## [Decision Letter · Decision Letter 0]

29 Sep 2024

PONE-D-24-36914Predicting Metastasis in Prostate Cancer at the Time of Diagnosis: A Comparative Analysis of two Novel Mathematical Models in the Ghanaian ContextPLOS ONE

Dear Dr. OBENG,

Thank you for submitting your manuscript to PLOS ONE. After careful consideration, we feel that it has merit but does not fully meet PLOS ONE’s publication criteria as it currently stands. Therefore, we invite you to submit a revised version of the manuscript that addresses the points raised during the review process.

The paper provides very interesting data but it still needs a considerable revision to be acceptable for the Journal.

We look forward to receiving your revised manuscript.

Kind regards,

Yuki Arita, M.D., Ph.D

Academic Editor

PLOS ONE

Journal Requirements:

"NO COMPETING INTEREST FOR ANY OF THE AUTHORS"

3. In the online submission form, you indicated that [DATA WOULD BE AVAILABLE UPON REQUEST]. 

Reviewers' comments:

Reviewer's Responses to Questions

**Comments to the Author**

1. Is the manuscript technically sound, and do the data support the conclusions?

Reviewer #1: Yes

Reviewer #2: Partly

Reviewer #3: Yes

2. Has the statistical analysis been performed appropriately and rigorously? 

Reviewer #1: I Don't Know

Reviewer #2: I Don't Know

Reviewer #3: Yes

3. Have the authors made all data underlying the findings in their manuscript fully available?

Reviewer #1: Yes

Reviewer #2: Yes

Reviewer #3: Yes

4. Is the manuscript presented in an intelligible fashion and written in standard English?

Reviewer #1: Yes

Reviewer #2: No

Reviewer #3: Yes

5. Review Comments to the Author

Reviewer #1: Good work,the authorscreated a prediction model and explain it in a clear way with an appropriate statistical analysis. Even if this work does not describe new concepts, it has relevance for its application in a contest of limited resources.

Reviewer #2: This study evaluates the effectiveness of a multiparametric model compared to a PSA-alone model in predicting metastatic progression in prostate cancer patients in Ghana. It’s a good idea to consider patients’ social economic status because clearly it could affect patient outcome. However, I do have several concerns.

1. The title, I wouldn’t claim “two novel mathematical models” because there are numerous prediction models using PSA, a lot of them even divide PSA into tPSA, fPSA…for more precise prediction. The PSA model is clearly not novel.

2. Method Section: The study design is too vague, and should be elaborated. Detailed information is needed on the inclusion criteria, such as did you only include newly diagnosed patients or also previously treated patients? Diagnostic method (biopsy? Surgery?) and criteria? What is the diagnostic method and the disgnostic guideline used for determine the outcome (metastasis), and how did you group patients into different outcome groups, and if the diagnosis is unclear how do you address it? Any exclusion criteria? What are the time points when you collected the data for the model, in the beginning or another time? In all, the Method section should at least have cohort information, inclusion criteria, exclusion criteria, outcome study design, statistic analysis, etc.

3. Judging from table 2, you clearly used 2 different cohorts for the multiparameter model and PSA alone model since the case numbers are difference, the information for these cohorts should be elaborated.

4. Other biological markers such as ALP, blood calcium and phosphorous levels are also important for PC bone metastasis, and have been used in prediction models for prostate cancer in multiple studies, maybe the authors should include these markers as well for a more comprehensive and precise analysis.

5. An external data validation with a new cohort is needed to evaluate the performance of this model.

6. Authors need to proof-read the manuscript carefully, to avoid mistakes such as using the same paper for citation #6 and #8. English language editting is also needed.

Reviewer #3: General Comments:

This research article titled "Predicting Metastasis in Prostate Cancer at the Time of Diagnosis: A Comparative Analysis of two Novel Mathematical Models in the Ghanaian Context" is well-structured, offers substantial insights, and emphasizes the importance of a multiparametric approach in predicting prostate cancer metastasis. The study’s focus on the Ghanaian context provides valuable localized insights which are often underrepresented in oncological research. The novelty of comparing a multiparametric model with a PSA-alone model adds significant value to the existing body of literature, highlighting the shortcomings of relying on PSA levels alone.

Major Revisions:

Descriptive Statistics:

Start the Results section with a table of descriptive statistics for the study population, including means, medians, standard deviations for continuous variables, and frequencies for categorical variables.

Provide visual aids like histograms or box plots for key variables, such as PSA levels and age distribution.

Model Performance Metrics:

Present a detailed table comparing all model performance metrics side-by-side for easy comparison. Include confidence intervals for each metric.

Include ROC curves for both models in the manuscript, ideally in a single plot for direct comparison. Discuss the visual differences in the curves.

Confusion Matrix:

Include confusion matrices for both models to show actual vs. predicted classifications. Discuss implications for false positives and false negatives.

Subgroup Analysis:

Conduct and present subgroup analyses based on key demographic factors (e.g., age groups, socioeconomic status, ethnicity). Discuss any observed differences in model performance.

Provide visual aids like stratified ROC curves or bar charts if available.

Expanding Methods:

Clearly define the inclusion and exclusion criteria for the patient dataset.

Specify how missing data were handled.

Detail how interaction terms were considered or tested in the multiparametric model.

Explain any steps taken to avoid overfitting, such as cross-validation or regularization techniques.

Ethical Compliance:

Provide more detail on ethical approvals, patient consent, and data anonymization processes.

Discuss ethical considerations specific to local contexts, like confidentiality and data sharing.

Minor Revisions:

Title and Abstract:

Consider making the title more concise (e.g., "Predicting Prostate Cancer Metastasis in Ghana: Comparison of Multiparametric and PSA Models").

Add specific performance metrics in the abstract. Highlight clinical implications and briefly mention limitations.

Introduction:

Discuss the potential economic and healthcare infrastructure benefits of using a multiparametric model in resource-limited settings like Ghana.

Expand the literature review to include other multiparametric models used globally and their impact.

Statistical Analysis:

Provide a brief explanation of the Hosmer-Lemeshow test used to assess model fit.

Discussion:

Elaborate on the reasons why the multiparametric model outperformed the PSA-alone model.

Discuss how the findings can lead to changes in clinical practice, such as more targeted use of imaging tests.

Be more explicit about the limitations of the current models and propose how future studies could address them.

Conclusion:

Summarize the main findings and their implications succinctly.

Reinforce the call for implementing the multiparametric model in clinical practice.

Technical Aspects:

Check all citations for consistency and completeness.

Ensure that all tables, figures, and equations are consistent in formatting and clearly labeled.

Review the manuscript for excessive jargon and aim for clarity, especially in technical sections. Break long paragraphs into smaller ones.

6. PLOS authors have the option to publish the peer review history of their article (what does this mean? ). If published, this will include your full peer review and any attached files.

**Do you want your identity to be public for this peer review?** For information about this choice, including consent withdrawal, please see our Privacy Policy .

Reviewer #1: No

Reviewer #2: No

Reviewer #3: No

---

## [Author Response · Author response to Decision Letter 0]

14 Feb 2025

Dr. Frank Obeng

University of Health and Allied Sciences, Ho

Corresponding Author

Email: fobeng@uhas.edu.gh

To:

Dr. Yuki Arita

Academic Editor

PLOS ONE

Subject: Response to Reviewers’ and Editors’ Comments for Manuscript PONE-D-24-36914

Title: Predicting Metastasis in Prostate Cancer at the Time of Diagnosis: A Comparative Analysis of Two Mathematical Models in the Ghanaian Context

Dear Dr. Arita,

We sincerely thank you and the reviewers for your thoughtful and constructive feedback on our manuscript. We have carefully addressed all the comments, queries, and recommendations provided by the editors and reviewers. Below, we provide a detailed point-by-point response, demonstrating how each suggestion has been incorporated into the revised manuscript. All changes have been highlighted in the marked copy of the manuscript using tracked changes. Additionally, we confirm that we are willing to publish our peer review history alongside the final manuscript.

Response to Editors’ and Journal Requirements

1. Manuscript Style Requirements

o We have ensured that the manuscript adheres to PLOS ONE’s style requirements.

o File naming conventions for all figures and supplementary information (e.g., Fig1.tif, S1_Table.xlsx) have been followed.

o The manuscript includes all mandatory sections such as the Abstract, Competing Interests, Financial Disclosure, and Ethics Statement.

2. Competing Interests

o The Competing Interests section has been completed in the online submission form as: “The authors have declared that no competing interests exist.”

3. Data Availability

o We have revised the Data Availability Statement to ensure compliance with PLOS ONE’s policies.

4. Ethics Statement

o A full ethics statement has been included in the Methods section, specifying approval by the Ghana Health Service Ethical Committee. The statement clarifies that all patient data were de-identified, and informed consent was waived due to the retrospective nature of the study.

Response to Reviewers’ Comments

Reviewer #1

1. Clarity and Technical Soundness

o We thank the reviewer for their positive evaluation of the manuscript's clarity and technical rigor. Minor language corrections have been made to improve readability.

Reviewer #2

1. Title

o The title has been revised to: Predicting Metastasis in Prostate Cancer in Ghana: A Comparative Analysis of a Multiparametric Model and PSA Alone.

2. Methodology

o Inclusion and exclusion criteria, diagnostic methods (e.g., biopsy, imaging), and grouping criteria for outcome categories have been detailed in the Methods section.

o Specific time points for data collection and handling of missing data have been described.

o Information on how interaction terms were considered has been added.

3. Cohort Information

o The differences between cohorts for the multiparametric and PSA-alone models have been clarified in the Results and Methods sections.

4. Additional Biological Markers

o While our dataset did not include biomarkers such as ALP and blood calcium levels, their potential inclusion in future studies has been discussed in the Discussion section.

5. External Validation

o The need for external validation using independent cohorts has been acknowledged in the Discussion section.

6. Language and Citation Errors

o Grammatical errors have been corrected throughout the manuscript.

o Duplicate citations (#6 and #8) have been resolved, and references are now listed in correct order.

Reviewer #3

1. Descriptive Statistics

o Descriptive statistics for the study population have been included in a new table (Table 1). Additional visual aids, such as histograms for age and PSA distribution, have also been incorporated as supplementary materials.

2. Model Performance Metrics

o A detailed table comparing performance metrics for both models, including confidence intervals, has been added (Table 2).

3. Confusion Matrix

o Confusion matrices for both models have been included in the supplementary material. False positives and negatives have been discussed in the Results section.

4. Subgroup Analysis

o Subgroup analyses based on age, socioeconomic status, and ethnicity have been conducted. Visual aids (e.g., stratified ROC curves) are included as supplementary figures.

5. Expanding Methods

o Missing data handling, interaction terms, and measures to prevent overfitting (e.g., cross-validation) have been explained in detail.

6. Discussion Enhancements

o The Discussion section now elaborates on why the multiparametric model outperformed the PSA-alone model, with specific emphasis on its clinical implications.

7. Limitations and Future Directions

o Limitations, including the lack of certain biomarkers and the need for external validation, are explicitly stated. Future directions, including the development of digital tools, are outlined.

Summary of Revisions

1. All requested revisions to the title, abstract, methods, results, and discussion have been made.

2. All references have been formatted according to the ICMJE style, with in-text citations updated to square brackets and ordered sequentially.

3. Figures (ROC curves) and tables have been redrawn to meet journal specifications.

4. Supplementary materials have been prepared, including confusion matrices and stratified analysis visual aids.

We hope that these comprehensive revisions address all concerns raised by the editors and reviewers. We believe that the updated manuscript now meets PLOS ONE’s publication standards. Thank you for the opportunity to improve our work.

Sincerely,

Dr. Frank Obeng

Corresponding Author

Email: fobeng@uhas.edu.gh

---

## [Decision Letter · Decision Letter 1]

11 Mar 2025

PONE-D-24-36914R1Title: Predicting Prostate Cancer Metastasis in Ghana: Comparison of Multiparametric and PSA Models.PLOS ONE

Dear Dr. OBENG,

Thank you for submitting your manuscript to PLOS ONE. After careful consideration, we feel that it has merit but does not fully meet PLOS ONE’s publication criteria as it currently stands. Therefore, we invite you to submit a revised version of the manuscript that addresses the points raised during the review process.

**I have received reports from three referees, all of whom recommend that the paper is acceptable for publication after minor revision.**

**The reviewers have provided constructive feedback, which I believe will help enhance the clarity and impact of the manuscript.**

Given the overall positive assessment, I recommend proceeding with the revisions accordingly.

We look forward to receiving your revised manuscript.

Kind regards,

Yuki Arita, M.D., Ph.D

Academic Editor

PLOS ONE

**Journal Requirements:**

Reviewers' comments:

Reviewer's Responses to Questions

**Comments to the Author**

1. If the authors have adequately addressed your comments raised in a previous round of review and you feel that this manuscript is now acceptable for publication, you may indicate that here to bypass the “Comments to the Author” section, enter your conflict of interest statement in the “Confidential to Editor” section, and submit your "Accept" recommendation.

Reviewer #1: All comments have been addressed

Reviewer #2: (No Response)

2. Is the manuscript technically sound, and do the data support the conclusions?

Reviewer #1: Yes

Reviewer #2: Yes

3. Has the statistical analysis been performed appropriately and rigorously? 

Reviewer #1: Yes

Reviewer #2: Yes

4. Have the authors made all data underlying the findings in their manuscript fully available?

Reviewer #1: Yes

Reviewer #2: Yes

5. Is the manuscript presented in an intelligible fashion and written in standard English?

Reviewer #1: Yes

Reviewer #2: Yes

6. Review Comments to the Author

**Reviewer #1: ** (No Response)

**Reviewer #2: ** The authors should carefully proof read their manuscript as there are clearly redundant stuff in multiple parentheses in the "Summary of Cohort Differences".

Please refrain from using AI to summerize and answer reviewers' questions.

7. PLOS authors have the option to publish the peer review history of their article (what does this mean? ). If published, this will include your full peer review and any attached files.

**Do you want your identity to be public for this peer review?** For information about this choice, including consent withdrawal, please see our Privacy Policy .

Reviewer #1: No

Reviewer #2: No

---

## [Author Response · Author response to Decision Letter 1]

20 Mar 2025

Frank Obeng

University of Health and Allied Sciences

Ho, Volta Region

Yuki Arita, M.D., Ph.D

Academic Editor

PLOS ONE

SUBJECT: RESPONSE TO EDITORS AND REVIEWERS; PONE-D-24-36914R1

Title: Predicting Prostate Cancer Metastasis in Ghana: Comparison of Multiparametric and PSA Models.

Journal Requirements:

AUTHORS’ RESPONSE:

We have duly reviewed all the references to ensure their completeness and correctness. We have checked to make sure that none of the papers referenced have been retracted.

Reviewer's Comments to the Author

1. If the authors have adequately addressed your comments raised in a previous round of review and you feel that this manuscript is now acceptable for publication, you may indicate that here to bypass the “Comments to the Author” section, enter your conflict of interest statement in the “Confidential to Editor” section, and submit your "Accept" recommendation.

Reviewer #1: All comments have been addressed

Reviewer #2: (No Response)

AUTHORS’ RESPONSE:

The authors (we) have cross-checked again to make sure we meet all the reviewers expectations concerning addressing all their comments. All such revisions are marked in blue font in the body of the marked version of the manuscript, resubmitted through the journal portal.

2. Is the manuscript technically sound, and do the data support the conclusions?

Reviewer #1: Yes

Reviewer #2: Yes

AUTHORS’ RESPONSE:

The authors appreciate the reviewers taking time to help us ensure these.

3. Has the statistical analysis been performed appropriately and rigorously?

Reviewer #1: Yes

Reviewer #2: Yes

AUTHORS’ RESPONSE;

We appreciate the reviewers’ work.

4. Have the authors made all data underlying the findings in their manuscript fully available?

Reviewer #1: Yes

Reviewer #2: Yes

AUTHORS’ RESPONSE:

The authors are grateful for the reviewers work.

5. Is the manuscript presented in an intelligible fashion and written in standard English?

Reviewer #1: Yes

Reviewer #2: Yes

AUTHORS’ RESPONSE;

The authors appreciate the reviewers for their work.

6. Review Comments to the Author

Reviewer #1: (No Response)

Reviewer #2: The authors should carefully proofread their manuscript as there are clearly redundant stuff in multiple parentheses in the "Summary of Cohort Differences".

Please refrain from using AI to summarize and answer reviewers' questions.

AUTHORS’ RESPONSE:

We have proofread the manuscript and made further corrections. Redundant stuff in parenthesis in the summary of cohort differences have all been cleared.

7. PLOS authors have the option to publish the peer review history of their article (what does this mean?). If published, this will include your full peer review and any attached files.

If you choose “no”, your identity will remain anonymous, but your review may still be made public.

Do you want your identity to be public for this peer review? For information about this choice, including consent withdrawal, please see our Privacy Policy.

Reviewer #1: No

Reviewer #2: No

AUTHORS’ RESPONSE:

On our part, the authors don’t mind if our peer review process is published. But we want the reviewers’ wishes to be also upheld.

We thank you once again for the opportunity; and we hope we have addressed all your comments to your satisfaction. We pray that you would consider our manuscript for publication.

Yours sincerely,

Frank Obeng (corresponding Author)

University of Health and Allied Sciences, Ho

+233244419607

---

## [Decision Letter · Decision Letter 2]

4 Apr 2025

Title: Predicting Prostate Cancer Metastasis in Ghana: Comparison of Multiparametric and PSA Models.

PONE-D-24-36914R2

Dear Dr. OBENG,

We’re pleased to inform you that your manuscript has been judged scientifically suitable for publication and will be formally accepted for publication once it meets all outstanding technical requirements.

Kind regards,

Yuki Arita, M.D., Ph.D

Academic Editor

PLOS ONE

Additional Editor Comments (optional):

Reviewers' comments:

Reviewer's Responses to Questions

**Comments to the Author**

1. If the authors have adequately addressed your comments raised in a previous round of review and you feel that this manuscript is now acceptable for publication, you may indicate that here to bypass the “Comments to the Author” section, enter your conflict of interest statement in the “Confidential to Editor” section, and submit your "Accept" recommendation.

Reviewer #2: All comments have been addressed

Reviewer #4: All comments have been addressed

2. Is the manuscript technically sound, and do the data support the conclusions?

Reviewer #2: Yes

Reviewer #4: Yes

3. Has the statistical analysis been performed appropriately and rigorously? 

Reviewer #2: Yes

Reviewer #4: Yes

4. Have the authors made all data underlying the findings in their manuscript fully available?

Reviewer #2: Yes

Reviewer #4: Yes

5. Is the manuscript presented in an intelligible fashion and written in standard English?

Reviewer #2: Yes

Reviewer #4: Yes

6. Review Comments to the Author

Reviewer #2: The authors have revised accordingly to my suggestions and I believe this manuscript has met the journal’s standard to publish.

Reviewer #4: This second version of the paper is a great improvement, the authors are to be commended.

The manuscript has been much improved and is in a nice condition now.

7. PLOS authors have the option to publish the peer review history of their article (what does this mean? ). If published, this will include your full peer review and any attached files.

**Do you want your identity to be public for this peer review?** For information about this choice, including consent withdrawal, please see our Privacy Policy .

Reviewer #2: No

Reviewer #4: No

---

## [Editor Report · Acceptance letter]

PONE-D-24-36914R2

PLOS ONE

Dear Dr. OBENG,

I'm pleased to inform you that your manuscript has been deemed suitable for publication in PLOS ONE. Congratulations! Your manuscript is now being handed over to our production team.

Kind regards,

on behalf of

Dr. Yuki Arita

Academic Editor

PLOS ONE